

# Optimising sampling of fish assemblages on intertidal reefs using remote underwater video

Katherine R. Erickson[1,2], Ana B. Bugnot[2,3] and Will F. Figueira[2,4]

[1] Centre for Marine Science and Innovation, University of New South Wales, Sydney, NSW, Australia
[2] School of Life and Environmental Sciences, University of Sydney, Sydney, NSW, Australia
[3] Oceans and Atmosphere, Commonwealth Scientific and Industrial Research Organisation, Brisbane, QLD, Australia
[4] Sydney Institute of Marine Science, Sydney, NSW, Australia

## ABSTRACT

**Background:** Assessing fish assemblages in subtidal and intertidal habitats is challenging due to the structural complexity of many of these systems. Trapping and collecting are regarded as optimal ways to sample these assemblages, but this method is costly and destructive, so researchers also use video techniques. Underwater visual census and baited remote underwater video stations are commonly used to characterise fish communities in these systems. More passive techniques such as remote underwater video (RUV) may be more appropriate for behavioural studies, or for comparing proximal habitats where the broad attraction caused by bait plumes could be an issue. However, data processing for RUVs can be time consuming and create processing bottlenecks.

**Methods:** Here, we identified the optimal subsampling method to assess fish assemblages on intertidal oyster reefs using RUV footage and bootstrapping techniques. We quantified how video subsampling effort and method (systematic *vs* random) affect the accuracy and precision of three different fish assemblage metrics; species richness and two proxies for the total abundance of fish, $MaxN_T$ and $MeanCount_T$, which have not been evaluated previously for complex intertidal habitats.

**Results:** Results suggest that $MaxN_T$ and species richness should be recorded in real time, whereas optimal sampling for $MeanCount_T$ is every 60 s. Systematic sampling proved to be more accurate and precise than random sampling. This study provides valuable methodology recommendations which are relevant for the use of RUV to assess fish assemblages in a variety of shallow intertidal habitats.

# INTRODUCTION

Accurate, rapid and robust quantitative techniques to assess fish communities are of importance to marine ecologists as they provide the basis for ongoing monitoring and assessment of community dynamics. This is especially important for systems in decline, such as shallow, complex estuarine habitats like oyster reefs, mangroves, seagrass beds, and

Corresponding author
Katherine R. Erickson,
k.erickson@unsw.edu.au

rocky reefs (*Gaylard, Waycott & Lavery, 2020*). They provide nursery habitat within the estuary (*Nagelkerken et al., 2015*) and support diverse fish assemblages that need to be monitored to maintain fishery and estuary health (*Whitfield & Elliott, 2002*). These habitats are also a focus for restoration, and optimising methodology for fish assessments is vital to maximise the uptake of monitoring and ensure restoration success (*Bosire et al., 2008*; *Baggett et al., 2015*).

In places such as marine reserves, non-invasive estimation methods are required over destructive catch estimations, which can bias future samples and compromise management objectives (*Andrew & Mapstone, 1987*; *Willis & Babcock, 2000*). Researchers thus rely on various non-invasive methods such as diver underwater visual census, netting, mark/recapture, and video (*Cappo, Harvey & Shortis, 2006*) to estimate population numbers.

Amongst these, baited (BRUV) and unbaited (RUV) remote underwater video systems are especially beneficial due to their suitability for use in areas exposed to strong currents, or waters too shallow or too deep for divers, and for the permanent record they supply (*Cappo, Harvey & Shortis, 2006*; *Langlois et al., 2010*). To date, most studies assessing the applicability of video techniques have focussed on baited videos (*Cappo, Harvey & Shortis, 2006*; *Langlois et al., 2010*; *Whitmarsh, Fairweather & Huveneers, 2017*; *Langlois et al., 2020*). However, unbaited RUVs are rising in popularity as they can be more appropriate for studies looking at fish behaviour, as they have little to no effect on the community they are recording (*Cappo, Harvey & Shortis, 2006*; *Mallet & Pelletier, 2014*). They are also better at capturing non-carnivorous species (*Goetze et al., 2015*) and for comparing community assemblages between proximal habitats, as it is impossible to know how far the attractant effect of the bait plume extends in the water column (*Asher, 2017*). However, it is recommended to employ more sampling effort for unbaited RUVs than baited, due to lower abundances of fish in the frame—longer soak times, or more replicates, to achieve the same level of statistical power (*Bernard & Götz, 2012*).

In addition, extracting data from the full videos can develop a 'bottleneck' in the analysis workflow due to observer fatigue, with 1 h of video taking up to 13 h to process (*Cappo, Harvey & Shortis, 2006*; *Campbell et al., 2015*). Recently, researchers have been recognising the value of time-saving applications of deep learning to automatically count and identify fish in videos (*Christin, Hervet & Lecomte, 2019*; *Ditria et al., 2021*; *Lopez-Marcano et al., 2021*). However, models do not perform as well in videos with complex backgrounds, poor visibility, differing light conditions, or cryptic camouflaged fish (*Salman et al., 2019*). Methods are being developed to cope with these conditions (*Salman et al., 2019*), but automated identification with greater than 70% accuracy is not yet possible for shallow habitats, for example oyster reefs, with all four of these issues. Additionally, each model must be trained with a dataset of thousands of annotated images of each species from the particular habitat (*Ditria et al., 2020*), which would be extremely time consuming and difficult, especially for less common species. Hence, it is key to identify the most efficient methods to manually count fish in videos like these to ensure accuracy.

One way to reduce processing time is by subsampling videos to assess fish communities. Studies have looked at how the number of RUV samples affects the accuracy of data

(*Garcia, Dias & Longo, 2021*), and examined optimal soak times (*Asher, 2017*; *Wong et al., 2019*; *Garcia, Dias & Longo, 2021*), but no studies have assessed precision and accuracy associated with subsampling fish metrics for RUVs.

The most common metrics used to assess fish assemblages from video are number of species captured across the duration of the video (hereafter, species richness) and MaxN, defined as the highest number of individual fish that appear in one frame (*Ellis, 1995*). It does not double count individual fish and thus does not result in overestimations (*Cappo, Harvey & Shortis, 2006*). According to the most recent review, it is the primary metric employed to assess fish communities through BRUV footage (81% of studies, *Whitmarsh, Fairweather & Huveneers, 2017*), and has a processing time:video length ratio of 0.5:1 (*Gladstone et al., 2012*). A new alternative metric, MeanCount, has only been used in 2% of BRUV studies but is rising in popularity (*Whitmarsh, Fairweather & Huveneers, 2017*). MeanCount uses either systematically or randomly selected individual frames from across the video to calculate a mean of the number of fish in the frame (*Conn, 2011*). Essentially, it is an occupancy-weighted MaxN and can be a more informative and useful metric to obtain. There is debate over which metric is more accurate, with one laboratory study showing that MeanCount is linearly related to true abundance, whereas MaxN was found to be hyperstable and underestimated true abundance (*Schobernd, Bacheler & Conn, 2014*). Other field and simulation studies have shown that it can be less precise than MaxN, and potentially over-inflate zero counts (*Stobart et al., 2015*; *Campbell et al., 2015*). Their relative value can change based on useage, as for RUVs in complex habitats there is high correlation between the two for structure-oriented species, but less so for mobile species (*Baker et al., 2022*).

Generally, previous studies have sampled species richness and MaxN across the duration of the video, while MeanCount is calculated from a subsample of frames ranging from 10–150 s (*Cappo, Harvey & Shortis, 2006*; *Schobernd, Bacheler & Conn, 2014*; *Follana-Berná et al., 2019*; *Cullen & Stevens, 2020*; *Kilfoil et al., 2021*). One study, however, has analysed the precision and accuracy associated with subsampling species richness. *Bacheler & Shertzer (2015)* found that subsampling every 30 s for 20-min BRUV footage results in 14% of species being missed. This study counted 210 species from 1,543 videos. Two studies have compared the precision and accuracy of BRUV data associated with MaxN and subsampling MeanCount. *Campbell et al. (2015)* found that MeanCount had lower precision than MaxN for eight economically important focal species, and recommended sampling MeanCount every 15 s, despite slight differences between species. *Bacheler & Shertzer (2015)* looked at three common focal species and advised sampling MeanCount every 50 s for each. The Gulf of Mexico where these studies were performed is very diverse, and therefore could have limited relevance to low abundance estuarine habitats, like oyster reefs, that have a smaller species richness. Additionally, these results might not be transferrable to unbaited RUVs, as BRUVs draw in more fish over time, whereas unbaited RUVs have a steadier abundance of fish. One simulation study indicated that for 3 h of RUV footage, sampling every 2 min was sufficient for accurate and precise MeanCount data, but this was done for just a single species (*Follana-Berná et al., 2019*; *Follana-Berná et al., 2020*). Methods for estimating MaxN and MeanCount should be

directly compared to decide on an optimal sampling strategy that maximises efficiency and robustness.

Moreover, frames for subsampling can be chosen systematically or randomly across the duration of the video. This has repercussions for the efficiency of data processing, as it could be logistically easier to sample MeanCount systematically (in real time) than randomly if the whole video is going to be watched regardless of other methods. No studies have experimentally compared the two sampling methods, but some articles have suggested using either (*Conn, 2011*; *Bacheler & Shertzer, 2015*). All metrics and both sampling methods should be explored when optimising efficiency as, contrary to previous recommendations, there could be a difference.

In this study we used RUV samples of fish assemblages from an intertidal habitat, oyster reefs, to (1) find the optimal soak time for RUVs to obtain accurate values of species richness in these systems, (2) compare the precision and accuracy associated with subsampling MaxN, MeanCount, and species richness using systematic and random methods, and (3) assess the trade-off between effort and information gained to make recommendations about the number of frames to sample.

## METHODS

Data collection for this study was carried out over two days in March 2019 at the Port Hacking oyster reefs, located in Sydney NSW (34°4′25″S, 151°7′7″E). Local species richness and true abundance of the fish community at this study site are not known. Video was collected with permit P03/0029-5.1 issued by the Department of Primary Industries, New South Wales, and under University of Sydney Ethics approval 2019/1571. Unbaited RUV footage was collected by setting out GoPro cameras (wide FOV, 1080 HD, 60 fps), attached to 50 cm × 50 cm × 30 cm metal frames, to film for 70 min over high tide around the reefs (See Fig. S1). Placement of cameras was split into three zones—Centre (interior of reef, 30 cm from edge), Edge (30 cm of reef edge and 30 cm of surrounding sand flat) and Off (1 m off the reef, facing towards empty sand). Visibility was consistent across all videos, and all fish visible were counted. Cameras were placed such that roughly 40% of the field of view was benthic habitat and 60% was water column. Ten of these videos, evenly spread over the three zones (Centre = 4, Edge = 3, Off = 3), were selected for the analysis presented here.

The first 5 min of each video was discarded to allow time for snorkelers to exit the area. All frames of the following 60 min of footage were fully annotated using EventMeasure software (SeaGis v. 5.12). Videos were watched in real time within the program and paused to record the species, and time, each individual fish entered and left the field of view (time in-time out, or TITO, from *Schobernd, Bacheler & Conn (2014)*). Hyper abundant fish (shoals of 200–800 fish) were handled differently as it was impossible to track every individual fish. The total time the shoal was in the frame was calculated, and then the number of fish in the frame was counted at five equal time intervals over this period. Species richness was derived from this dataset, as well as MaxN and MeanCount for each species. The average processing-time:video length ratio for all videos was 8:1.

Optimal 'soak time', defined as length of time the camera is deployed and filming, was investigated through species richness. The cumulative number of species seen was taken from TITO data at 1-min intervals across the 60 min of footage for all 10 videos (following *Cappo, Harvey & Shortis, 2006*; *Schramm et al., 2020*). A GAM curve and 95% CI were fitted using the *ggplot2* package (v3.3.2; *Wickham, 2016*) to aid visual interpretation.

The TITO data was used to create a dataset of the number of fish present for every second of the video using a custom Microsoft Excel (Microsoft, Redmond, WA, USA) script (3,600 s-points in total per video). For each of the 10 videos, MaxN, MeanCount, and species richness were estimated by bootstrap sampling this dataset with two subsampling methods, systematic and random, over a series of levels of effort. Both methods used sampling intensities of 360, 180, 120, 60, 30, 20, 15 and 12 sampling points over the 3,600-s length of the video. These estimated values were then compared to the 'true' values, where 'true' refers to the accurate values for each metric obtained from the TITO dataset, and not the true abundance or richness of the study site (which is unknown).

For systematic sampling, points were distributed at equal intervals through the video (*i.e.*, for 12 sampling points, a frame was taken every 300 s over the course of the video). The need for equal sample intervals restricted the number of unique subsets available for each video. The highest sample size of 360 points required a sampling interval of 10 s, where there are only ten different possible options within a 60 min (3,600 s) video. Where there were more than ten possible datasets for a given sample size, the 'rand' function in Microsoft Excel (Microsoft, Redmond, WA, USA) was used to pick a random selection of ten of these. For random sampling, points were drawn at random from all possible points, with replacement, with the process repeated 10 times using the *boot* package (v.1.3-27; *Davison & Hinkley, 1997*; *Canty & Ripley, 2021*) in R version 1.4.1717 (*R Core Team, 2019*).

For these two methods, accuracy and precision curves were calculated for each video and the point of diminishing returns was found by averaging over all ten videos. Precision was defined as the coefficient of variation (CV) within each sample size. Accuracy was defined as the percentage absolute deviation of the observed estimate from the 'true' value (as calculated from TITO data).

$$Deviation = \frac{|Truth - Estimated|}{Truth} * 100$$

Our unbaited videos had low abundance and many zeros, which is to be expected for unbaited videos on reefs with rare, cryptobenthic species. While this is not an issue for community analyses, it does make species-level accuracy and precision estimates highly variable, and hence difficult to evaluate (See Fig. 1 for examples of three species, the common mobile species yellowfin bream *Acanthopagrus australis*, the common juvenile tarwhine *Rhabdosargus sarba*, and the cryptobenthic goby *Redigobius macrostoma*). For this reason, we have conducted the accuracy and precision analyses of our abundance proxies by pooling across species to give the metrics total MaxN (MaxN$_T$) and total MeanCount (MeanCount$_T$). Previous studies on fish abundance have combined metrics

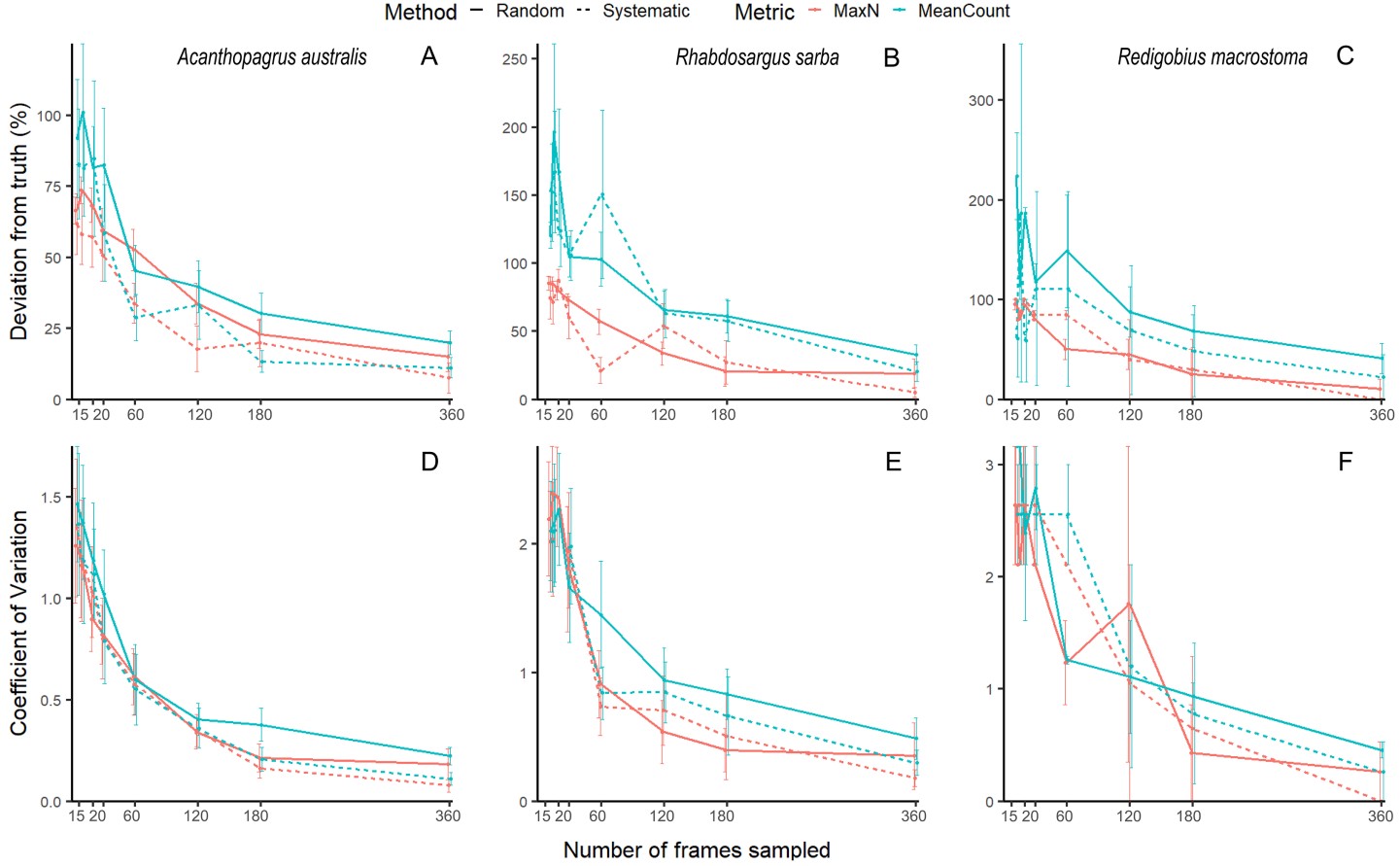

**Figure 1** Mean (±SE) accuracy and precision of MaxN and MeanCount for three individual species. *Acanthopagrus australis* accuracy (A) and precision (D), *Rhabdosargus sarba* accuracy (B) and precision (E), and *Redigobius macrostoma* accuracy (C) and precision (F).

like MaxN into 'total abundance' to facilitate analysis (*Scott et al., 2015*; *Bond et al., 2018*; *Piggott et al., 2020*; *Smith et al., 2021*).

We compared the accuracy and precision of each method and metric by testing for effects of metric, method and sampling effort on CV and deviation, using generalised linear models in the package *glmmTMB* (*Brooks et al., 2017*) in R 4.0.2. Two schooling species *Ambassis jacksoniensis* and *Atherinomorus vaigiensis* obscured other patterns present in the data, therefore analysis of CV and deviation (not species richness, as this metric was not affected) was repeated with the schooling species removed. Interactions between sampling effort, metric and method were tested, and interpreted using *post-hoc* comparisons in the package *emmeans* (*Lenth et al., 2018*).

To identify the optimal sampling point, beyond which increasing effort resulted in minimal improvements to accuracy and precision, 95% confidence intervals were calculated for each sampling effort to identify where the errors differed significantly from zero—aka the accurate values obtained from the TITO dataset. However, the lower bounds were not negative when comparing to zero, even at small sample sizes. Therefore, optimal values were assessed visually, based on the rate of curve deceleration.
| Order | Family | Species |
|---|---|---|
| **Table 1 List of species found in the videos.** | | |
| Atheriniformes | Atherinidae | *Atherinomorus vaigiensis* |
| Beloniformes | Belonidae | *Tylosurus gavialoides* |
| Mugiloformes | Mugilidae | *Mugil cephalus* |
| Perciformes | Ambassidae | *Ambassis jacksoniensis* |
| | Blenniidae | *Omobranchus anolius* |
| | | *Omobranchus rotundiceps* |
| | Gerreidae | *Gerres subfasciatus* |
| | Girrellidae | *Girella tricuspidata* |
| | Gobiidae | *Arenigobius bifrenatus* |
| | | *Cryptocentroides gobioides* |
| | | *Favonigobius exquisitus* |
| | | *Redigobius macrostoma* |
| | Sillaginidae | *Sillaginodes punctatus* |
| | | *Sillago ciliata* |
| | Sparidae | *Acanthopagrus australis* |
| | | *Rhabdosargus sarba* |

## RESULTS

### Soak time

A total of 16 different species from 10 different families were observed over the course of the analysis (Table 1). Species richness increased over the duration of the video, with the greatest difference between the first frame and 15 min (0–5.5 species), after which the curve began to plateau and gradually increase up to the 60 min mark (5.5–7.5 species) (Fig. 2).

### Deviation from truth (Accuracy)

All three metrics generally had lower deviation, with smaller standard error, as sampling effort increased, but the effect of the random and systematic methods differed between the metrics.

For the all-species analysis (Fig. 3A), the systematic method had significantly lower deviation than the random (Table 2). There was a significant interaction between sampling effort and method (Table 2). *Post-hoc* analysis showed that $MeanCount_T$ had lower deviation than $MaxN_T$ and Species Richness, while there was no difference between $MaxN_T$ and Species Richness (Table S1). The differences were more pronounced at low sampling efforts (Table S1).

Species Richness and $MaxN_T$ showed similar patterns across sampling effort—sampling from 12 frames to 120 frames resulted in a sharp 30% drop in deviation and, beyond this, the slope of the curve plateaus yielding a 10% drop in deviation from 60 to 360 frames. For systematic $MeanCount_T$, the metric and method with lowest deviation, increasing

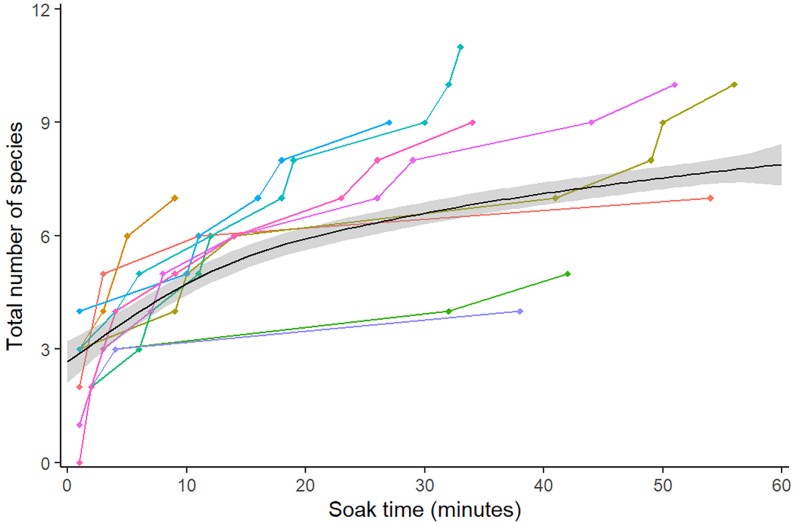

**Figure 2 Mean species accumulation curve over the duration of all 10 videos.** Colour represents the 10 different videos. A GAM was fitted to the points to assist visual interpretation, grey band shows 95% confidence interval.                                                     

sampling from 12 frames to 60 frames resulted in a steep 23% drop in deviation. Further increases in sampling effort to 360 frames decreased deviation more gradually by 7%.

When schooling species were removed from the analysis, the accuracy for $MeanCount_T$ improved between 5-10% across all sampling efforts, whereas $MaxN_T$ did not improve (Fig. 3B). The systematic method had significantly lower deviation than the random (Table 2). There was a significant interaction between sampling effort and metric (Table 2). *Post-hoc* analysis showed that $MeanCount_T$ was always more accurate than $MaxN_T$, but this difference was more pronounced at low sampling efforts (Table S2).

For systematic $MeanCount_T$, increasing sampling from 12 frames to 60 frames returned a 14% drop in deviation and, beyond this, the curve flattens yielding a 5% drop in deviation from 60 to 360 frames.

## Coefficient of variation (Precision)

All three metrics generally had a lower coefficient of variation (CV), with smaller standard error, as sampling effort increased, but the effect of the random and systematic methods differed between the metrics.

For the all-species analysis (Fig. 3C) there was a significant interaction between sampling effort and metric, and between sampling effort and method (Table 2). *Post-hoc* analysis shows that there was no difference between the random and systematic methods for any metric, however there was a strong trend where the systematic method had lower CV than the random for $MeanCount_T$ at high sampling efforts (360, 180, 120) (Table S3). Systematic $MeanCount_T$ had lower CV than the other approaches at high sampling efforts (360, 180) (Table S3). Species richness behaved differently to the abundance metrics as effort decreased, the CV did not rise as sharply (Fig. 3C).

For systematic $MeanCount_T$, the metric and method with the lowest CV, increasing sampling from 12 frames to 120 frames results in CV dropping steeply from 0.44 to 0.05

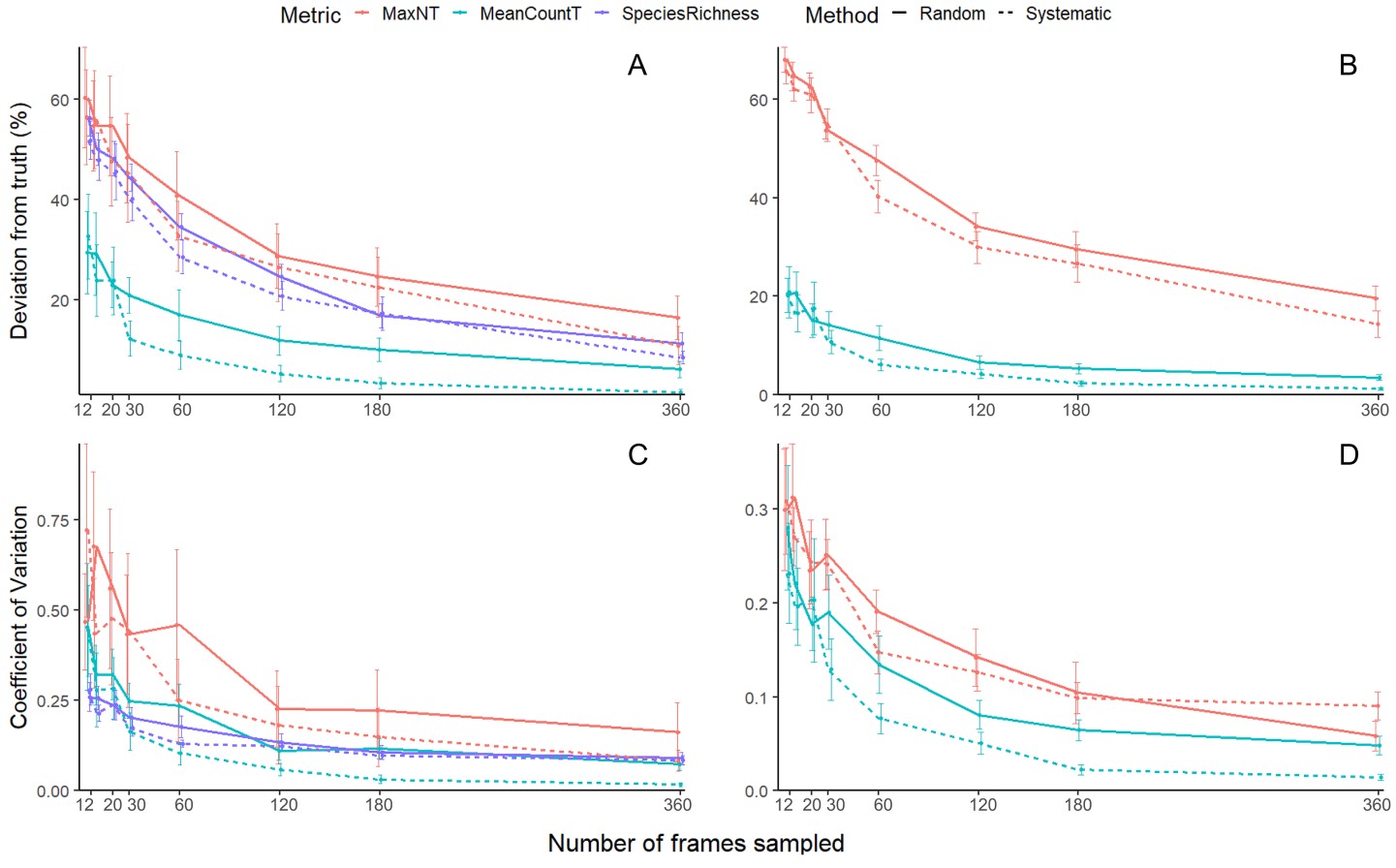

**Figure 3 Mean (±SE) accuracy and precision of species richness, MaxN$_T$ and MeanCount$_T$.** All species accuracy (A) with shoaling species removed (B), all species precision (C) with shoaling species removed (D).

and, beyond this, the curve flattens yielding a 0.04 drop in deviation from 120 to 360 frames (Fig. 3C). When schooling species were removed from the analysis, the CV improved ~50% across all sampling efforts (Fig. 3D). There was a significant interaction found between sampling effort, metric and method (Table 2). *Post-hoc* analysis shows that at low sampling efforts, there was no difference between the metrics or methods (Table S4). At high sampling efforts, MeanCount$_T$ was more accurate than MaxN$_T$, and the systematic method was more accurate than random for MeanCount$_T$ only (Table S4). The curve of the systematic MeanCount$_T$ approach did not have a clear inflexion point (Fig. 3D).

# DISCUSSION

Robust methodologies for monitoring fish assemblages are of critical importance for maintaining the health, and improving the restoration success, of declining estuarine habitats such as oyster reefs (*Whitfield & Elliott, 2002*; *Baggett et al., 2015*; *Gaylard, Waycott & Lavery, 2020*). Unbaited videos are the most appropriate technique for these habitats as they overcome scientific and ethical issues associated with destructive or invasive sampling (such as trapping, UVCs or BRUVs) that can stress fish and influence their behaviour and home range (*Cappo, Harvey & Shortis, 2006*; *Mallet & Pelletier, 2014*;

**Table 2  Results of generalised linear models for the effects of sampling effort, metric and method on deviation (accuracy) and coefficient of variation (precision).** Bold values denote statistical significance at the $p < 0.05$ level.

| Variable | $\chi^2$ | Df | $p$ value |
|---|---|---|---|
| Deviation from truth—all species | | | |
| i | 191.487 | 1 | **<0.001** |
| Method | 5.145 | 1 | **0.023** |
| Metric | 135.847 | 2 | **<0.001** |
| i:Method | 0.016 | 1 | 0.899 |
| i:Metric | 12.397 | 2 | **0.002** |
| Method:Metric | 0.090 | 2 | 0.956 |
| i:Method:Metric | 0.141 | 2 | 0.932 |
| Deviation from truth—shoaling species removed | | | |
| i | 321.399 | 1 | **<0.001** |
| Method | 5.017 | 1 | **0.025** |
| Metric | 867.704 | 1 | **<0.001** |
| i:Method | 0.337 | 1 | 0.562 |
| i:Metric | 78.062 | 1 | **<0.001** |
| Method:Metric | 0.154 | 1 | 0.695 |
| i:Method:Metric | 0.070 | 1 | 0.791 |
| Coefficient of variation—all species | | | |
| i | 147.451 | 1 | **<0.001** |
| Method | 8.501 | 1 | **0.004** |
| Metric | 55.475 | 2 | **<0.001** |
| i:Method | 6.133 | 1 | **0.013** |
| i:Metric | 13.953 | 2 | **0.001** |
| Method:Metric | 3.519 | 2 | 0.172 |
| i:Method:Metric | 4.264 | 2 | 0.119 |
| Coefficient of variation—shoaling species removed | | | |
| i | 206.075 | 1 | **<0.001** |
| Method | 5.760 | 1 | **0.016** |
| Metric | 37.941 | 1 | **<0.001** |
| i:Method | 1.668 | 1 | 0.197 |
| i:Metric | 11.779 | 1 | **<0.001** |
| Method:Metric | 5.845 | 1 | **0.016** |
| i:Method:Metric | 11.243 | 1 | **<0.001** |

*Asher, 2017*). This study provides useful insights into the differences in trade-offs between effort and accuracy/precision for subsampling three fish community metrics, extracted from unbaited underwater video focussed on characterising fish assemblages on intertidal oyster reefs. While all three metrics were more accurate and precise as sampling intensity increased, they each behaved differently across the sampling intervals. Subsampling MeanCount$_T$ tended to be more accurate and precise than MaxN$_T$ or species richness, but

differences in precision faded at low sampling efforts. Accuracy and precision tended to increase when using the systematic subsampling method (over the random) and when removing schooling species. We did not compare $MaxN_T$ and $MeanCount_T$ directly, or comment on the accuracy of any of the metrics, as the true abundance and local richness of the study site is unknown.

The species accumulation curve began to plateau at 15 min, at the point of 75% species detection, matching the findings of a previous study on RUVs (*Asher, 2017*). Species saturation was not achieved by 60 min, due to the difficulties of spotting the less common goby and blenny species such as *Redigobius macrostoma* and *Omobranchus rotundiceps*. Other methods such as UVC may be more appropriate for accurately recording the abundance of cryptobenthic species (*Watson et al., 2005*); however, these methods are not usually possible in the shallow water of intertidal habitats such as oyster reefs. In the videos, these species camouflage into the background, and are far less abundant than oyster blennies (*Omobranchus anolius)* or bream (*Acanthopagrus australis*). This would suggest, noting the large range of values over the ten videos, that the optimal soak time for a general question regarding abundant or fishery-dependant species, excluding cryptobenthics, would be 15 min. A total of 60 min or more would be appropriate for studies looking at less abundant species. This is similar to recommendations of 60–90 min for BRUVS in seagrass, where the complexity of habitat and cryptic behaviour of species requires a longer soak time (*Gladstone et al., 2012*). Estimates for species richness steadily became more accurate and precise until a sample size of 120, with more modest benefits after this. If deviation is to be kept under 10% and CV under 0.1 (in order to find less common species, for instance camouflaged blennies and gobies), then we recommend not subsampling videos to capture species richness. In a study focussed on abundant species, 120 subsamples would minimise error to 20%. The random method tended to be less accurate, but not significantly so, therefore we recommend systematic (sampling every 30 s) for ease of workflow. *Bacheler & Shertzer's (2015)* study on BRUVs found a similar result for the comparatively lower effort of 50 subsamples. This is likely due to unbaited cameras having a lower probability of filming less common species than baited, so RUVs need more effort to capture true species richness. Additionally, their study was based in the Gulf of Mexico, with a much higher overall species richness (210 *vs* 16) which is likely to affect the sampling effort needed.

Overall, subsampling $MaxN_T$ led to lower precision and accuracy than subsampling $MeanCount_T$. For example, at 60 subsamples, there is 9% variation from the 'true' TITO value for $MeanCount_T$ and 32% for $MaxN_T$. Even at the highest subsampling effort (360 frames), deviation for $MaxN_T$ already reaches 16%, which is why this study recommends recording $MaxN_T$ in real time. No previous studies have attempted to subsample MaxN, but our result is reasonable in the context of $MaxN_T$ being a single value for the whole video—the real value may only occur once or twice. $MeanCount_T$ instead is an average of multiple values across the video and so is more robust to variation caused by the subsampling.

Systematic video subsampling was more accurate than random for the $MeanCount_T$ metric (Table 2). We suggest the reason random video subsampling was less accurate is

because of the temporally correlated nature of fish presence. Some of the bootstrap replicates from random sampling were entirely in the first or second half of the video, thus missing large groups of fish that occurred, and resulting in higher variation. Systematic sampling has a higher chance of catching these groups of fish that are only in the frame for a few minutes. This pattern can be seen in the $MaxN_T$ accuracy curve, though not as strongly, as $MaxN_T$ is one value rather than an average of values over time, and thus is less affected by temporal correlation. Therefore this study recommends systematically sampling videos for $MeanCount_T$, rather than randomly. It was not previously known whether systematic or random sampling had lower error, and it was recommended to use either (*Bacheler & Shertzer, 2015*). This recommendation is convenient as systematic sampling is logistically simpler, being more easily accommodated in existing laboratory video analysis work flow, and corresponds to methods in previous articles (*Conn, 2011*; *Bacheler et al., 2017*). Considering the higher accuracy and precision of systematic sampling for the metric of $MeanCount_T$, we can recommend subsampling videos to calculate this metric to reduce processing time. The 'inflexion point' on the curve for accuracy of all species (Fig. 3A) is at 60 frames, where reducing sampling effort further results in a sharp decrease in accuracy, and an increase in variability (larger standard error bars). Analysis that excluded schooling species (Fig. 3B) had higher precision for both $MaxN_T$ and $MeanCount_T$, and higher accuracy for $MeanCount_T$. Excluding these species increases accuracy overall, as the small chance of missing the schooling species results in a huge error (*i.e.*, getting a $MeanCount_T$ of 3, instead of 250). The 'inflexion point' on the curve for accuracy is between 30 and 60 frames, but the variability (standard error bar) is higher for 30 frames. The pattern is similar for precision, where reducing effort to less than 60 frames results in a sharp decrease in precision and an increase in variability (larger standard error bars). To minimise effort, and keep error low at ~6% and CV at 0.1, we recommend sampling every 60 frames (see Table 3 for a summary).

Due to the true values being obtained with a TITO approach, the cost in time per frame cannot be calculated for each of the sampling intensities conducted here. However, the recommended method for video analysis was tested in a later study on oyster reefs (C. Pine, 2020, personal communication) which found that sampling every 60 s for $MeanCount_T$, simultaneously recording $MaxN_T$ and species richness, resulted in a reduction of processing time:video length ratio from 8:1 to 1:1, with only a small percentage increase in error. The metrics of $MaxN_T$ and species richness had an even lower ratio of 0.5:1, as the entire video can be watched at 2× speed for RUV data in these low abundance habitats. A recent study on RUVs suggested using Frequency of Occurrence, a presence/absence metric derived from species richness, in situations where a quick and robust assessment of fish assemblage composition is required (*Baker et al., 2022*). Researchers could consider this method over subsampling when fast data processing is required.

Accuracy and precision estimates are typically directly related to the relative frequency of occurrence of the subject of the monitoring (*Lechene et al., 2019*). It would thus be ideal to estimate sampling effort needed for individual species within this dataset. Unfortunately, this was not possible at the species level in this study given the low

**Table 3 Summary of recommendations for subsampling unbaited RUV in shallow intertidal habitats.**

| Metric | Recommendation |
|---|---|
| Soak time | 15 min for abundant sp. <br> >60 min for rare sp. |
| Species richness | Every 20 s for abundant sp. <br> Real time (no subsampling) for rare sp. |
| $MaxN_T$ | Real time (no subsampling) |
| $MeanCount_T$ | Every 60 s |
| Method | Systematic better than random |

abundance within our RUV footage, as it was unbaited in a complex habitat with multiple rare, cryptobenthic species. The high frequency of zero-values resulted in erratic accuracy and precision curves (see Fig. 1) at the species level. Therefore, our recommendations are based on proxies of total fish abundance for all species, and researchers interested in accuracy and precision for individual species, especially the more uncommon species, would expect to need higher effort. Subsampling these species would result in larger errors, and we caution researchers to account for the uncertainty arising from this in any estimations and expectations calculated from this methodology.

The low abundance and richness of these videos means that our recommendations are applicable for researchers studying unbaited videos in similar low abundance habitats, such as estuarine mangroves, rocky reefs, seagrass and bivalve reefs. Rapid and robust assessments of fish assemblages in these important habitats will not only assist in developing management strategies to monitor and maintain the health of our estuaries (*Whitfield & Elliott, 2002*), but are also necessary for successful restoration by helping to define goals and evaluate the progress of projects (*Bosire et al., 2008*; *Baggett et al., 2015*).

## CONCLUSIONS

Estimating the true abundance of fish species is of critical importance for the monitoring and management of fish communities. However, data processing time constraints are a significant limitation for using video surveys to estimate fish abundance. Therefore, knowing precise, accurate and efficient ways to subsample this data is essential. This study has generated guidelines for subsampling unbaited remote underwater video on oyster reefs, or other similarly low abundance complex intertidal habitats, for three different community metrics. In summary, we recommend a workflow where the video is watched in real time (or potentially at 2× speed) to estimate species richness and total MaxN ($MaxN_T$), while total MeanCount ($MeanCount_T$) is estimated every 60 s. This reduces the data processing effort from 8 h per hour of footage down to 1 h, while maintaining low errors in accuracy and precision. These guidelines will assist researchers to evaluate fish community dynamics quickly and efficiently, thus promoting successful monitoring and restoration of threatened estuarine habitats like oyster reefs, mangroves and seagrasses.

## ACKNOWLEDGEMENTS

We specially thank the researchers, students and volunteers involved in the fieldwork.

### Funding

This project was funded by an Australian Research Council Linkage Project LP180100732 with partners Sydney Institute of Marine Science, Department of Primary Industries, and the Nature Conservancy. The funders had no role in study design, data collection and analysis, decision to publish, or preparation of the manuscript.

### Grant Disclosures

The following grant information was disclosed by the authors:
Australian Research Council Linkage Project: LP180100732.
Sydney Institute of Marine Science.
Department of Primary Industries.
Nature Conservancy.

### Competing Interests

The authors declare that they have no competing interests.

### Author Contributions

- Katherine R. Erickson conceived and designed the experiments, performed the experiments, analyzed the data, prepared figures and/or tables, authored or reviewed drafts of the article, and approved the final draft.
- Ana B. Bugnot analyzed the data, authored or reviewed drafts of the article, and approved the final draft.
- Will F. Figueira conceived and designed the experiments, performed the experiments, analyzed the data, authored or reviewed drafts of the article, and approved the final draft.

### Animal Ethics

The following information was supplied relating to ethical approvals (*i.e.*, approving body and any reference numbers):

The University of Sydney provided full ethics approval (2019/1571).

### Field Study Permissions

The following information was supplied relating to field study approvals (*i.e.*, approving body and any reference numbers):

Field collections were authorised by permit P03/0029-5.1 issued by the Department of Primary Industries.

### Data Availability

The raw data from fish videos is available in the Supplemental Files.

## Supplemental Information

Supplemental information for this article can be found online at http://dx.doi.org/10.7717/peerj.15426#supplemental-information.

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
