# Peer review of "Optimising sampling of fish assemblages on intertidal reefs using remote underwater video"

_PeerJ, doi:10.7717/peerj.15426_

## Round 0.1 · original submission · Major Revisions

Dear Dr. Erickson,

Sorry for the delay and sent your review before but was very hard to find reviewers. Now, I am sending the positive reviewer's comments that I believe increase the quality of your study.

All the best

Juan Pablo

Reviewer 1 ·

Basic reporting

The paper presents a comparison between methods to subsample underwater videos aiming to estimate fish richness and abundance. The text is well written, clear and concise. The analysis is very interesting. However, I made some comments requesting more details regarding the methods.
The main issue that is confusing about the study is that MaxN and MeanCount are estimates of abundance for each species. However, I quote: "An attempt was made to subsample MeanCount and MaxN metrics for individual species, but due to the extremely low abundance resulting from RUV data on oyster reefs, the high presence of zero values prevented such species-specific analysis. Therefore, species abundances were combined into metrics of total fish species, denoted as MaxNT and MeanCountT". If MaxN or MeanCount are used to estimate abundance for each species, I do not understand what MaxN total and MeanCount total mean. All species combined? I am not sure if I understand the point here. Maybe give more details about this and add more references to support this choice.
I understand the importance of developing methods to subsample long videos in order to decrease the time to analyze them. However, I consider it important to discuss the relevance of refining these methods even when automated fish identification and counting is being developed.

Experimental design

The study is within the aims and scope of the journal. The research question is well defined, relevant and meaningful. However, I have some considerations about the method.
Line 129 - Please, explain it better the concept of point overlay in EventMeasure and TITO. If possible, add a reference for the use of TITO, or give more details why to use this measure, and why it is considered the truth.
I would also like more details of MaxNT. What is the practical use of this? Is there a reference for this choice?

Validity of the findings

All underlying data have been provided; they are robust, statistically sound, and controlled. Most of my considerations are in general comments.

Additional comments

Line 124 - I suggest including a picture of the hardware used, if possible.
Table 1 - Usually, in cases like this, the list of species is on the supplementary material. However, this list is relatively short, so it also fits in the manuscript, in my opinion. However it would be also interesting to present this list with columns of each method, showing which species were recorded by each method. If the authors choose to do this, I would include only in the supplementary material. But this is completely up to the authors.
Line 132 - For each fish species or each individual?
Line 247 - "MeanCount tended to be more accurate and precise than MaxN or species richness, but differences in precision faded at low sampling efforts." MaxN total or MaxN? And what do you mean? How is it possible to compare abundance measures with richness?
Line 275 - "...but studies have shown that MeanCount is more accurate and precise than sampling MaxN in real time without subsampling (Schobernd et al. 2014)." In which cases? RUVs? BRUVs? Please give more details and references for this statement. And please explain why MaxN is much more used in studies and supported by the literature than MeanCount, even though MeanCount, according to this sentence, is more accurate.
Line 323-325 - "The high frequency of zero-values led to wildly fluctuating accuracy and precision curves. However, the outcomes based on the entire assemblage will still inform sampling strategies seeking to inform about the entire assemblage. They will also serve to temper expectations about accuracy and precision of estimates on individual taxa/species derived from similar sampling.In my opinion, it is not accurate and highly speculative to infer this. Be very careful to state that. I would delete this sentence. I reckon it is not possible to extrapolate this result to MaxN by species because it would very likely be full of zeros in all studies anyway. For that reason, I do not see the point of doing a total Maxn.
Line 328 - "True abundances are impossible to measure due to time constraints and data processing bottlenecks". Be careful with sentences like this. Indeed it is very difficult to have true abundances of species in nature, that is the reason why we use ESTIMATES. Subsample is to facilitate the ESTIMATIVE, not to make it possible. Change this sentence, please.
Line 332 - Please, make it clearer that you are presenting results from MaxN total and MeanCount Total here and along the whole text.
Line 333 - In the last sentence, I understood what you meant, but the sentence should be rewritten to make it clearer to the reader. What does it mean? What is the importance of this?

·

Basic reporting

The study compares two different metrics (MaxN and Mean Count) and methods (random and non-random subsampling) when estimating accuracy and precision of fish abundance and richness from RUVs on oyster reefs. I think this could be a valid contribution to the methods field of using video to study fish assemblages. However, there are several aspects that needs to improve. The English need to be carefully revised. Some of the references are wrongly cited, which needs to be corrected. The wording need to be revised for clarity, so as not to overstate findings, or imply things that were not done. The figures need to improve. The raw data is provided, and the authors respond to their research questions.
I think the manuscript would improve with re-wording of several points in the discussion.

Experimental design

I miss a description of local species richness at the study site, as well as true abundance. If this is not known from previous studies, this should be stated, since these metrics would influence how these results are applicable in other biogeographic provinces with differing species richness and abundance. I also miss a discussion of how applicable these results are in studies elsewhere, considering these metrics (since species richness and abundance were very low).

Validity of the findings

Data seem to be robust to my knowledge. However, I feel that the results are not fully supporting the conclusions due to the issues mentioned in section 1. and 2. Several of the findings need careful re-wording so as not to overstate findings.

Additional comments

Introduction

I am not sure that RUVs (or any video system) would be accurate when surveying small and cryptic species. Watson, D.L., Harvey, E.S., Anderson, M.J. et al. 2005 (A comparison of temperate reef fish assemblages recorded by three underwater stereo-video techniques. Marine Biology 148) concluded that no small cryptic species (e.g. blennids and gobids) were recorded by any of the video techniques. In general, video techniques are considered less effective when surveying cryptic species so I would be careful with this statement (e.g. Goetze et al 2019, Methods in Ecology & Evolution, DOI: 10.1111/2041-210X.13189; Pita et al 2013, Marine & Freshwater Research 65).

Line 65: The paper by French et al actually says the contrary: videos were about the least effective in detecting small and cryptic species. Please be careful with your citations.

Line 70: This statement need a reference.

Line 76: The novelty: subsampling RUVs

Line 85: Used only in 2% of studies with BRUVs, not all studies including videos (according to the stated reference). The first half of the sentence is also directly copied from Whitfield et al (2017). It could be good to state the disadvantages with mean count as well.

Line 88-89: It would be good to define/explain how/in what way MeanCount would be a more accurate method, since this is the focus on the paper (see Schobernd et al 2014 for example).

Line 93: Revise English

Line 97-103: Regarding time and ideal MeanCount this should also be related to local species richness (compare for example the Pacific vs Mediterranean). A study concerning a single species would therefore be difficult to compare to a study on the whole fish assemblage. What about species accumulation curves comparing these metrics across biogeographic provinces? (see for example Cornell and Harrison 2014, Ann Rev Ecol, Evol & System. 45, Garcia et al 2021, Journ. of Fish Biol 99). Were any metrics like this taken into account when comparing these different recommendations for MeanCount?

Line 101: Unclear what “more than 90 frames” means? Maybe a language error?

Line 109: “..regardless of other..”

Line 109- 111: Unclear meaning, revise sentence

Line 112: “..from an intertidal..”


Methods
How were samples standardized with regard to area/volume where fish were sampled? I.e., how far from the camera were fish surveyed?

Results

Line 201-204: Revise English

Discussion

Line 256: How do you define “rare” species? In this context, blennies may not be rare, but your sampling method is not ideal to detect them. This leads to the idea that blennies are “rare” while instead the method fails to detect them. I suggest to clarify this.

Line 261: Is it even possible to detect and accurately survey cryptic species with this method? (see previously suggested literature).

Line 276: This was not what the study by Schobernd et al (2014) found. MaxN tended to be nonlinearly related to true abundance in the Schobernd study, but the authors also caution against using MeanCount until more studies are performed. They did not evaluate accuracy and precision as this study did. The Schobernd study used simulations and aquaria experiment where true abundance was known. This is another thing that I miss in this study, is true abundance and species richness known? Or is this study the first to estimate this? Despite this study is mainly comparing these different metrics, I think this should be clarified, and wording carefully revised to not overstate findings.

Line 286: Suggest change “better” to “lower”

Line 293: Aren’t error and standard error bars the same thing?

Line 295: I would suggest using “schooling fish” instead of hyperabundant - i.e. a large number of fish appearing in a short period of time/few frames. If they were constantly hyperabundant, the probability to miss them if subsampling is used would be minimal.

Line 298-99: Revise English

Line 308: This may be true for this particular study, but maybe not on coral reefs with high species abundance and diversity.

Line 323-325: Revise English

Line 325-326: A bit unclear what this means

Conclusions

These recommendations may be true for the study site in this study, but how do they apply to different settings with higher fish abundance and species richness?



Figures

Figure 2: Numbers on axes needs to be larger font sizes to be easily read. The language in the legend should be revised. It is a bit difficult to see the standard deviation bars, would suggest to make them bolder to be able to compare them easier (if this is not a problem with the resolution of my version).

Tables

Table 2: Check consistency for p-values

---

## Round 0.2 · Minor Revisions

Dear Authors,

Both reviewers considered that your manuscript can be accepted after of include all the comments mentioned in the reviews. Thus, I invite you to review carefully these comments and submit this new version.

All the best,

Juan Pablo

Reviewer 1 ·

Basic reporting

The manuscript has substantially improved after the first revision. The references were updated, the text was refined and the analysis was clarified.

Experimental design

The relevance of the research question and the method applied got more highlighted after the review.

Validity of the findings

All underlying data have been provided; they are robust, statistically sound, & controlled.

Additional comments

Line 90 - I think something is missing here? “As of 2017…”

·

Basic reporting

This paper uses Remote Underwater Video (RUV) techniques on intertidal oyster reefs to how video subsampling, effort, and method affect the accuracy and precision of fish assemblages richness and abundance. I have read the manuscript and the response to reviews. I believe the authors have done a good job responding to the previous set of reviews – which I consider essential at this state of the peer review process. This is an interesting paper that provide the optimal subsampling method to assess fish assemblages and valuable recommendations for the use of RUV in low-abundance intertidal reefs. Therefore, I think the manuscript should be published if the authors go over some minor revisions. Below I show some of these:

1. I agree with the authors that it is important to optimize the sampling of fish assemblages in order to decrease the data processing effort. However, I also consider it important to include in the introduction section more ecological evidence based on what is the importance of considering assessing fish assemblages in low-abundance intertidal reefs in comparison with other similar environments? Are there broader considerations of how changes in habitat conditions can affect the fish behavior on intertidal reefs?

2. As previously mentioned by reviewer 2, video techniques are often not adequate when assessing cryptobenthic species, even using an underwater visual census strongly underestimates the presence of cryptobenthic species. Recent studies demonstrated that cryptobenthic reef fishes have a disproportional influence on coral reef trophodynamics and functional ecology (Brandl et al. 2019, Science). Accounting for them could affect the optimal soak time and the accuracy and precision metrics of abundance and species richness, therefore, have you considered it convenient to exclude this type of species when RUV techniques are applied to assess these fish assemblages.

3. In the methods section, I suggest highlighting the importance of how the URVs techniques may improve or complement the assessment of the fish assemblages in comparison with other underwater “traditional” strategies.

4. Also, I think that the discussion needs more explanation about the ecological implications of the results observed on the use of RUV techniques in the assessment of fish assemblages in the study area.

5. I suggest a straightforward conclusion that includes the ecological implication of these assessments on intertidal habitats on a regional or global scale. Also, I would expect a conclusion based on ecological evidence: what is the importance of consider assess fish assemblages in low-abundance intertidal reefs? In my opinion this information is relevant as it can help explain the observed patterns of species richness and abundance associated with subsampling compared to the patterns observed in nature.

Experimental design

The methods section is well-defined. Sampling occurred are different sections of the reef, which make a robust design. The text is clear and easy to understand. Minor comments below.

Validity of the findings

The study provides valuable insights which are relevant to the use of RUVs to assess fish assemblages on intertidal reefs. However, I my opinion is necessary to have previous information on the species potentially present in the area to clarify whether the richness recorded in this study is similar to that observed in other studies or if the presence-absence of the species is influenced by other aspects such as seasonal variation or the level of habitat condition.

Additional comments

Background

1. Line 27: should be “…these methods are costly and destructive…”

Introduction

2. Line 59-60: this statement needs some references to support it.
3. Line 65: Maybe better insert “In addition,” before “Due to the lower abundances”

Methods

4. Line 142: delete “from permit” before “P03/0029-5.1”.
5. Line 144: replace “GoPros” with “GoPro cameras”.
6. Line 169 replace “ten” with “10”.
7. Line 175 replace “sampling points” with “points”.

Results

8. Line 219: replace “begins” with “began”.

Discussion

9. Line 309: insert “instead,” after MeanCountT.
10. Line 311-313: This statement needs a reference.

Conclusions

11. Line 360: replace “abundances” with “abundance”.

---

## Round 0.3 · accepted · Accept

Dear authors,

Thanks for submitting the new version of your manuscript. I reviewed the reviewer's comments and I believe that your manuscript can be accepted for publication. I believe that this study will be an important resource to marine biologists and ecologists that use remote videos to estimate biodiversity metrics.

All the best

Juan Pablo